# Cutting-Edge Therapies and Novel Strategies for Acute Intermittent Porphyria: Step-by-Step towards the Solution

**DOI:** 10.3390/biomedicines10030648

**Published:** 2022-03-11

**Authors:** Miriam Longo, Erika Paolini, Marica Meroni, Paola Dongiovanni

**Affiliations:** 1General Medicine and Metabolic Diseases, Fondazione IRCCS Ca’ Granda Ospedale Maggiore Policlinico, Pad. Granelli, Via F Sforza 35, 20122 Milan, Italy; longo.miriam92@gmail.com (M.L.); eryka.paolini.93@gmail.com (E.P.); maricameroni11@gmail.com (M.M.); 2Department of Clinical Sciences and Community Health, Università degli Studi di Milano, 20122 Milan, Italy; 3Department of Pharmacological and Biomolecular Sciences, Università degli Studi di Milano, 20133 Milan, Italy

**Keywords:** AIP, PBGD, heme, liver metabolism, α-lipoic acid, insulin

## Abstract

Acute intermittent porphyria (AIP) is an autosomal dominant disease caused by the hepatic deficiency of porphobilinogen deaminase (PBGD) and the slowdown of heme biosynthesis. AIP symptomatology includes life-threatening, acute neurovisceral or neuropsychiatric attacks manifesting in response to precipitating factors. The latter promote the upregulation of 5-aminolevulinic acid synthase-1 (ALAS1), the first enzyme of heme biosynthesis, which promotes the overload of neurotoxic porphyrin precursors. Hemin or glucose infusions are the first-line therapies for the reduction of ALAS1 levels in patients with mild to severe AIP, while liver transplantation is the only curative treatment for refractory patients. Recently, the RNA-interference against ALAS1 was approved as a treatment for adult and adolescent patients with AIP. These emerging therapies aim to substitute dysfunctional PBGD with adeno-associated vectors for genome editing, human PBGD mRNA encapsulated in lipid nanoparticles, or PBGD protein linked to apolipoprotein A1. Finally, the impairment of glucose metabolism linked to insulin resistance, and mitochondrial aberrations during AIP pathophysiology provided new therapeutic targets. Therefore, the use of liver-targeted insulin and insulin-mimetics such as α-lipoic acid may be useful for overcoming metabolic dysfunction in these subjects. Herein, the present review aims to provide an overview of AIP pathophysiology and management, focusing on conventional and recent therapeutical approaches.

## 1. Introduction

Porphyrias are a group of rare genetic disorders caused by inborn errors in one of the eight enzymes involved in heme biosynthesis, and they lead to a reduction in heme availability alongside the overproduction and accumulation of harmful heme precursors (porphyrins). Although heme synthesis occurs in all cell types, 80% and 15% of its biogenesis arises from erythrocytes and hepatocytes, respectively [1]. Indeed, porphyrias are categorized into two different groups depending on the principal site of porphyrin accumulation: the erythropoietic cutaneous porphyrias (ECPs) and the acute hepatic porphyrias (AHPs). The latter includes δ-aminolevulinic acid (ALA) dehydratase deficiency porphyria (ADP), acute intermittent porphyria (AIP), variegate porphyria (VP), and hereditary coproporphyria (HCP) [2,3].

AIP is the most common and severe form of acute porphyria with hepatic implications, and results from a deficiency of about 50% in the *hydroxymethylbilane synthase* (*HMBS*) gene, which encodes the third enzyme of the heme pathway, referred to as porphobilinogen deaminase (PBGD). AIP’s clinical manifestations include life-threatening, acute neurovisceral and psychiatric attacks precipitated by metabolic, hormonal, and environmental factors such as fasting, infections, drugs, alcohol, or physical stress. Precipitating factors increase the demand for hepatic heme production, which normally exerts negative feedback on 5-aminolevulinic acid synthase-1 (ALAS1), the first rate-limiting enzyme of the heme pathway. Nonetheless, the deficiency in the free heme pool caused by the partial loss of PBGD activity leads to upregulation of ALAS1, thereby promoting the excess of the heme neurotoxic metabolites 5-aminolevulinic acid (ALA) and porphobilinogen (PBG), which accumulate in porphyric livers, plasma, and urine. The most frequent symptoms reported during the crisis are abdominal pain, nausea, vomiting, weakness, and constipation, and the crisis is often accompanied by sympathetic nervous system discomfort [4]. Peripheral motor neuropathy, complications of the central nervous system, and other chronic conditions, including hypertension, hyponatremia, and kidney disorders, may arise in patients with severe AIP, which represents a growing life-long burden for patients and their relatives [4,5,6].

Diagnosis of AIP is frequently underestimated and disease surveillance aims to educate at-risk *HMBS*-mutation carriers to avoid exposure to precipitating factors. Management and/or prophylaxis of acute episodes include the administration of hemin as the first-line approach and carbohydrate loading in cases of mild attacks and an absence of hemin. Both therapies induce the downregulation of hepatic *ALAS1*, thus alleviating symptoms and reducing porphyrin levels. However, the only curative option is through the replenishment of the defective PBGD to the extent that refractory AIP recipients who underwent liver transplantation (LT) have shown a complete biochemical and clinical remission [7,8,9,10].

The major steps forward for the treatment of AIP were carried out in recent years with (1) the approval of givosiran (Alnylam Pharmaceuticals, Inc., Cambridge, MA, USA), which inhibits ALAS1 through RNA interference (RNAi) in patients aged at least 12 years, and (2) with the ongoing efforts in the development of therapies based on the replacement of nonfunctional PBGD with hepatic delivery of the mRNA or protein [11,12,13,14,15]. Interestingly, the characterization of metabolic profiles in AIP experimental models and humans has highlighted alterations in glucose and lipid metabolism, insulin resistance (IR), and hepatic mitochondrial dysfunction, which offers the possibility of considering insulin and insulin-mimetics for the clinical management of AIP [16,17,18].

## 2. The Link between Penetrance, Prevalence, and Genetic Traits in AIP

AIP is an autosomal dominant inherited disorder with a low penetrance and significant heterogeneity of mutations in the *HMBS* gene [19], located on the 11q24.1–q24.2 chromosome. Currently, over 400 *HMBS* mutations have been recognized in the AIP scenario, which leads to the loss of PBGD enzymatic activity (Human Gene Mutation Database HGMD, http://www.hgmd.cf.ac.uk/ac/index.php) (accessed on 10 February 2022). Alternatively, spliced transcript variants encode two different isoforms of the *HMBS* gene: the erythroid-specific one is encoded by exons 2–15 and its promoter is positioned at intron 1, while the housekeeping isoform is encoded by exon 1 and exons 3–15 with the housekeeping promoter located in the 5’ flanking region upstream of exon 1 [20]. The majority of patients with AIP carry the *HMBS* mutations in exons 3–15, which affects both PBGD isoforms, while mutations in exon 1 do not influence PBGD functionality in erythrocytes [20]. Additionally, defects in the 5′-promoter region of the *HMBS* gene have been detected in individuals with AIP. Regarding the 400 mutations in the *HMBS* gene, there are 31 CpG dinucleotides in the 1086 base-pair coding sequence that are considered mutable as a consequence of the oxidative deamination of methylated cytosines [21,22]. Although AIP is a low-penetrance disorder, it has been identified that few *HMBS* mutations are relatively common, such as the p.R173W and p.R167Q variants, caused by CpG methylation, and the p.G111R and p.W198X variants, which are most frequent in Argentina and Sweden due to the founder effect [23,24,25].

To unravel the phenotypic expression and transmission of AIP, Lenglet and colleagues exploited the Exome Variant Server (EVS) database to estimate the prevalence and penetrance of deleterious *HMBS* mutations in the general population and performed an intra-familial study that included 253 families from the French reference center for porphyria (CFP). Although the prevalence of AIP genetic traits was high (1/1299), they estimated its penetrance at 0.5–1% in the general population. Conversely, in French families, they identified 496 subjects who were symptomatic, and 1672 relatives who were asymptomatic *HMBS* carriers, rating the penetrance of AIP at 22.9%. Moreover, the EVS database recognized 127 *HMBS* nucleotide variants from subjects of Caucasian and African families. These genetic variations included 22 heterozygous missense mutations, 12 synonymous mutations, 11 untranslated region (UTR) variants, and 82 intron variants, among which 20 missense mutations appeared to be harmful in the descendants of French patients with AIP. Particularly, the p.Ala122Pro, p.Arg167Gln, p.Arg175Gln, p.Arg195Cys, and p.Arg355Gln variants encoded an inactive protein, while the p.Glu86Val, p.Arg321His, and p.Asp359Asn mutations reduced PBGD enzymatic activity to 50%. The presence of null alleles, leading to the complete loss of function of the PBGD protein, was more frequent than missense mutations in families with AIP and correlated with the severity of disease manifestation [26]. An intrafamilial study identified a significant phenotypic correlation—defined as the manifestation of a symptomatic trait—among siblings, which may be explained by genetics, with a probability of 73%. This correlation decreased from first to third degree relatives, suggesting that the outcome of the disease was even influenced by other genetic factors. In addition, the phenotypic correlation was higher between kinship members of the same age and decreased between different generations, indicating that even environmental factors could modulate the risk of acute episodes and influence AIP inheritance [26]. Accordingly, a 3-year prospective study of symptomatic patients with AIP in 11 European countries revealed that the annual incidence of symptomatic manifestations was around 0.13 per million, while its prevalence was 5.9 per million in Europe [27]. The discrepancy in the prevalence of *HMBS* mutations and the occurrence of acute manifestations, coupled with an estimated penetrance of 20–50% in families with AIP versus 1% for the general population, strongly emphasize the role of environmental factors in triggering AIP attacks [26,27].

As we previously mentioned, AIP is an autosomal dominant disorder with the same mutational distribution among men and women, even if the manifestation of acute attacks affects more females than males. Accordingly, Baumann and colleagues found that in Finland, women displayed an AIP penetrance of 41% [28].

The majority of *HMBS* mutations encompassing p.W198X, c.1073delA, and p.R26C were correlated with both higher penetrance and clinical manifestations, while other variants such as p.R167W, p.R225G, and c.G33T were associated with lower penetrance and mild clinical events. Concerning the p.R173W mutation, a reduced penetrance was observed in Finland, whereas it was much higher in Northern Sweden and Spain [29]. A correlation between the presence of p.R116W, p.R173W, p.R149X, p.Q217H, p.G218R, p.A219P, and p.A330P mutations and the severity of AIP manifestations was also predicted through a bio-informatics approach [30]. To conclude, AIP prevalence in the population is high, although its penetrance is exceptionally low, resulting in the difficult identification of asymptomatic individuals with AIP. Moreover, the different penetrance rates of symptomatology in the relatives of French patients with AIP underline that AIP inheritance could be modulated by the environment and other genetic factors independently of *HMBS* variations.

## 3. Heme Biosynthesis and AIP Pathogenesis

The first step in the heme pathway begins in the mitochondria, where glycine and succinyl-coenzyme A (succinyl-coA), derived from the Krebs cycle, are converted into ALA through the ALAS enzyme [2]. ALAS exists in isoforms which are encoded by two separate genes: the ubiquitously expressed ALAS1, and the erythroid-specific isoform ALAS2. Two molecules of ALA are transported into the cytoplasm and are condensed by ALA dehydratase (ALAD), producing monopyrrole-PBG. In the third step of heme biosynthesis, four molecules of PBG are combined by PBGD enzyme to form hydroxymethylbilane (HMB), which is converted to uroporphyrinogen (UROgen) III by UROgen III synthase (UROS). Due to its molecular instability, HMBS may spontaneously form into a tetrapyrrolic ring (UROgen I) when the UROS enzyme is absent.

Then, UROgen III decarboxylase (UROD) removes the carboxyl groups from the side chains of uroporphyrinogen III to obtain coproporphyrinogen (COPROgen) III. At this step, COPROgen III is shuttled into the mitochondria, where COPROgen III oxidase (CPOX) catalyzes two sequential steps of oxidative decarboxylation, thereby forming protoporphyrinogen (PROTOgen) IX. Subsequently, PROTOgen oxidase (PPOX) removes six atoms of hydrogen from the PROTOgen IX molecule, producing protoporphyrin (PROTO) IX, the first colored tetrapyrrole intermediate. Finally, the insertion of a ferrous ion (Fe^2+^) into the PROTO IX ring by ferrochelatase (FECH) allows the final production of heme [31].

This molecule is incorporated into hemoproteins (e.g., cytochrome (CYP) P450 and hemoglobin), thereby participating in a wide array of intrahepatic and extrahepatic functions, including mitochondrial oxidative phosphorylation (OXPHOS), drug detoxification, and systemic oxygen transport.

AIP is caused by genetic mutations affecting PBGD activity with the consequent depletion of heme production in the liver. The main consequence of the shutdown of heme biosynthesis is the loss of negative feedback on ALAS1, which leads to the overproduction and accumulation of the porphyrin precursors ALA and PBG in the liver and systemic circulation. The excess of ALA is the major trigger of neurological damage by inducing autonomic and peripheral neuropathy and also encephalopathy. Due to its structural similarity with the γ-aminobutyric acid (GABA) and glutamate neurotransmitters, ALA may modify the GABAergic system by acting as a GABA-receptor agonist and by inhibiting GABA release from the pre-synaptic terminals [6,32]. Moreover, ALA participates in the production of free radicals and reactive oxygen species (ROS) and may promote hepatic ROS-induced genomic and mitochondrial DNA damage (mtDNA) that predisposes patients with AIP to a higher risk of developing hepatocellular carcinoma (HCC) [33,34,35]. Although it has been broadly established that the symptoms of acute attacks are mainly attributed to the accumulation of neurotoxic metabolites, there is no conclusive evidence to date about the factors influencing penetrance and the wide variability in clinical manifestations. Indeed, ~90% of patients with AIP, many of whom are asymptomatic, high excreters (ASHE) of porphyrins, remain asymptomatic throughout their life, suggesting that the pathophysiology of acute events may not be directly triggered by the accumulation of neurotoxic byproducts.

Metabolic alterations were observed in both animal models and patients with AIP and suggest new insights into the pivotal role of the liver in predisposing the pathogenesis of acute attacks. The first evidence was obtained from patients with AIP who improved biochemical and clinical abnormalities after LT [36,37]. Conversely, patients with HCC who received livers from donors with AIP developed neurovisceral symptoms together with increases in ALA and PBG levels [10].

Most recently, it emerged that AIP experimental models show impaired glucose metabolism and mitochondrial respiration capacity during acute attacks, while carriers of AIP displayed high body weight, hyperinsulinemia, and alterations in serum lipid profiles combined with IR [16,38,39]. These findings suggest that AIP may be considered a genetic disorder characterized by an umbrella of metabolic dysfunctions, thereby adding new additional knowledge about its pathophysiology and, possibly, novel curative strategies. The effect of *PBGD* haploinsufficiency on heme biosynthesis is represented in Figure 1.

Therefore, in this section we describe the metabolic alterations that occur during stressful conditions such as fasting, focusing on the most recent findings investigating glucose metabolism and mitochondrial failure.

### 3.1. Impaired Glucose Homeostasis and IR Contribute to Acute Attacks of AIP

Several studies underlined that caloric impoverishment exasperates acute attacks of AIP and many individuals with AIP with severe symptomatology maintain an inadequate carbohydrate intake, suggesting that dietary carbohydrates or glucose infusions may be supplemented to reduce the severity of these events [38,39,40,41,42]. Matkovic and colleagues demonstrated that the hormonal regulation of carbohydrates may be modeled by porphyrinogenic drugs such as 2-allyl-2-isopropylacetamide (AIA) and 3,5-diethoxycarbonyl-1,4-dihydrocollidine (DDC). By treating female rats with AIA and DCC, they found that phosphoenolpyruvate carboxykinase (PEPCK) and glycogen phosphorylase (GP) activities, which are involved in gluconeogenesis and glycogenolysis, respectively, were around 43–46% lower than those of controls [43]. Then, Collantes et al. investigated hepatic glucose homeostasis and metabolism in a genetic model of AIP bearing around 30% of the activity of hepatic PBGD. After 14 h fasting, the AIP mice showed an increased expression of ALAS1, which prompted the release of urinary PBG, whereas ALA secretion appeared stable. Notably, while wild-type (Wt) mice induced glycogenolysis in response to caloric restriction with the consequent increment of systemic glucose precursors as lactate, pyruvate, and alanine, the AIP mice were unable to use glycogen storages and early triggered gluconeogenesis and ketogenesis [41]. In the liver, the degradation of glycogen is catalyzed by the glycogen phosphorylase isoenzyme, which is stimulated by glucagon, while insulin promotes its inactivation. In both the inducible and genetic models of AIP, serum insulin exceeded the physiological range by up to six-fold, while glucagon levels were reduced, supporting the notion that hormonal dysregulation may perturb the hepatic regulation of glucose metabolism, thus contributing to the inhibition of gluconeogenesis and glycogenolysis [41,43]. The imbalance of insulin and glucagon in the serum was amenable to PBGD deficiency since PBGD–liver-gene therapy was able to re-establish hepatic glycogenolysis. Disturbances in glucose metabolism associated with PBGD deficiency were also recently revealed by our group. We found that PBGD-mRNA silencing in human hepatoma cells (HepG2), especially during fasting stimulus, reduced the expression of glycolytic enzymes such as glucokinase (GCKR), phosphofructokinase (PFK) and pyruvate kinase (PK), suggesting that the downregulation of heme biosynthesis may even affect glycolysis [17].

Evidence from the literature shows that hepatic PBGD depletion is even correlated with a delayed glucose peak in the glucose tolerance test (GTT), supporting the notion that peripheral IR may participate in AIP pathogenesis [44]. In a case-control study, 44 Spanish patients with AIP and 55 controls were enrolled and stratified according to age, gender, and body mass index (BMI). It was observed that the insulin resistance index (HOMA-IR) was significantly higher in eight individuals with AIP (18.2%), among which five were associated with obesity (BMI > 30). Interestingly, patients characterized by IR and high levels of serum insulin tended to have stable disease, suggesting an achievable protective role for IR against the acute attacks [16]. A population-based study conducted by Lithner and colleagues reported that the onset of diabetes mellitus not only induced a complete biochemical remission of the attacks in patients with AIP, but it also counteracted the HCC development, possibly by normalizing ALA levels [38,45]. Nonetheless, it was reported that diabetes may activate hepatic ALAS1 in mice, and a case report revealed that the presence of concomitant overt diabetes mellitus in one patient with AIP was paralleled by the late manifestation of acute episodes, indicating that the impact of IR in AIP pathology needs to be further elucidated [46,47].

Finally, a metabolomic approach provided several markers of glycolysis and energy-conversion pathways encompassing acetate, citrate and pyruvate which resulted higher in the urines of 73 asymptomatic *HMBS* carriers, thus suggesting that the metabolic reprogramming may occur in the AIP pathogenesis, even in the absence of overt symptomatology. Interestingly, high concentrations of urinary acetate, citrate, and pyruvate appeared to be disease-specific and allowed discrimination of individuals with AIP from those affected by other types of porphyria, such as porphyria cutanea tarda (PCT) [48]. In another study, the serum metabolic profiles of patients with AIP denoted 15 variables which could discriminate between asymptomatic patients with AIP and healthy subjects. Among them, serum analysis by liquid chromatography revealed that levels of several lipid species, including sphingomyelins (C16:0, C24:0, and C24:1) and phosphatidylcholines (C32:1, C36:1, and C36:3), were significantly higher in the serum of patients with AIP [39]. Conversely, branched-chain amino acids (BCAA: valine, leucine, isoleucine) and aromatic amino acids (AAA: tryptophan, tyrosine) were significantly lower in patients with AIP. BCAAs and AAAs are precursors of protein synthesis, hormones, and neurotransmitters, and their reduction could increase the risk of liver cirrhosis and neurological disorders in patients with AIP [48]. Thus, such studies have identified plasma amino acids, lipid species, and glycolytic and energy-conversion metabolites associated with AIP serum profiles as possible biomarkers of IR and disease progression.

### 3.2. The Occurrence of Mitochondrial Failure in AIP Pathophysiology

Several findings underlined the involvement of glucose metabolism, IR, and mitochondrial turnover in AIP pathophysiology, but their contribution to the manifestation of acute episodes has not been elucidated yet. Notably, heme biosynthesis is essential for respiratory activity, indicating a possible link between its depletion and mitochondrial dysfunction in AIP scenarios [49]. Heme is partially synthesized in mitochondria and is regulated through negative feedback on the ALAS enzyme, which catalyzes the mitochondrial assembly of ALA by using glycine and succinyl-CoA derived from the tricarboxylic acid (TCA) cycle. ALAS1 activity depends on the TCA cycle for its substrates, such as succinyl-CoA, while the erythroid ALAS2 interacts directly through a C-terminal-specific binding domain with the β-subunit of succinyl-CoA synthetase-2 [50]. Heme is a crucial cofactor for OXPHOS components, such as ubiquinol-cytochrome-c oxidoreductase (complex III), cytochrome oxidase (complex IV), and cytochrome c, which transport electrons from complex III to complex IV. Moreover, the occurrence of AIP episodes due to environmental factors such as infections, fasting, and the use of drugs such as phenobarbital (PB), require hepatic hemoproteins, including mitochondrial respiratory cytochromes and the CYP-450 enzymes, for xenobiotic detoxification [51]. It was demonstrated that the administration of ALA in HepG2 cells triggered oxidative damage in mitochondrial and nuclear DNA, causing mitochondrial membrane hyperpolarization along with increased ROS production and decreased ATP content [33,34,35]. In vivo, rodents deficient in *PBGD* displayed insufficient levels of NADH, FADH2, succinyl-CoA, and ATP during PB-induced acute attacks, thereby modifying the activity of the respiratory chain [52,53]. In particular, the administration of PB in AIP mice did not affect hepatic complex IV or cytochrome c, while mitochondrial complexes I and III were significantly compromised. These complexes contain iron–sulfur clusters that represent the targets of oxidative stress caused by exacerbated ALA production. Furthermore, even complex-II activity was affected by PB administration, resulting in the maladaptation of the TCA cycle [52]. In addition, *HMBS* deficiency in a knock-in-mouse model that was biallelic for the *Hmbs* c.500G > A (p.R167Q) mutation significantly decreased the enzymatic activity of complexes III and IV in the hippocampus, mainly due to the reduction in heme availability. Consequently, the lower respiration capacity reduced the mitochondrial membrane potential, leading to higher ATP consumption rather than production [54].

Furthermore, the fasting stimulus in AIP mice prompted hepatic gluconeogenesis and ketogenesis rather than glycogenolysis, suggesting that carbohydrate deprivation was able to impair mitochondrial bioenergetics with a consequent increment of ketone bodies in the circulation and in the liver in order to provide energy [41]. Herrick and colleagues disclosed that the circulating level of lactate after glucose overload was higher in six patients with AIP than in six healthy subjects, along with pyruvate, thus postulating that an increased amount of lactate could be associated with heme reduction which, in turn, affected the cytochromes of the mitochondrial respiratory chain [42]. Accordingly, the increased systemic rate of lactate was correlated with the failure of mitochondrial OXPHOS in ALA-treated rats, together with decreased superoxide dismutase and citrate synthase enzymatic activities in the liver [55]. The impairment of mitochondrial bioenergetics could even be influenced by the unbalancing of the mitochondrial lifecycle and turnover, including fusion, fission, and mitophagy, which leads to alterations in mitochondrial dynamics, mass, shape, and metabolism. It was observed in both AIP mice and humans that the derangement of mitochondrial biogenesis was correlated with aberrations in mitochondrial morphology and functionality [17,52,54,56]. The analysis of the hepatic transcriptome of AIP mice, after PB exposure, indicated that the majority of differentially expressed genes belong to mitochondrial biogenesis and OXPHOS, both regulated by the peroxisome proliferator-activated receptor gamma coactivator 1-alpha (PPARGC1A, encoding the PGC1-α transcription factor). The upregulation of mitochondrial enzymes, such as ATP synthase F1 subunit alpha (Atp5a1), ATP synthase F1 subunit gamma (Atp5c1), and ATP synthase peripheral stalk-membrane subunit b (Atp5f1), in AIP mice indicated that these organelles played a critical role during acute manifestations [57].

Little evidence has highlighted the alterations of mitochondrial morphology in AIP pathophysiology. The hepatic parenchyma of mice in which porphyria was induced through a diet containing griseofulvin displayed an enlargement of mitochondrial shape parallel to the presence of para-crystalline inclusions and disorientated cristae, possibly due to the unbalancing of the mitochondrial lifecycle [58,59]. Interestingly, intramitochondrial crystalline inclusions, which were characterized by parallel filamentous structures of 12 nm in diameter, possibly due to mitochondrial cristae derangement, were observed in the liver biopsies of subjects affected by PCT [60].

Aberrations in mitochondrial shape and size may be caused by defects in mitochondrial dynamics and metabolism, which are both modulated by PGC1-α. It was demonstrated that the fasting stimulus in AIP mice triggered the expression of hepatic ALAS1 via PGC1α, underlining its implication in precipitating acute episodes of AIP [2,17,54,61]. Our group investigated metabolic dysfunction in response to PBGD downregulation in hepatocytes, focusing on mitochondrial failure in terms of bioenergetics sources and dynamics. The activity of PGC1α was prompted by fasting, whereas the levels of optic atrophy 1 (OPA1) and mitofusin 2 (MFN2), which orchestrate the mitochondrial fusion of inner and outer membranes, respectively, were downregulated. Conversely, dynamin-1-like protein (DRP1), which is involved in mitochondrial fission and OXPHOS activity, was upregulated. Furthermore, the protein levels of complexes I and IV of the respiratory chain appeared reduced, indicating that fasting could switch the mitochondrial dynamics towards fission rather than fusion, thereby promoting OXPHOS failure and ATP shortage [17]. Most recently, it was demonstrated that a deficiency of FETCH, a gene involved in the last step of heme production and responsible for the development of erythropoietic protoporphyria (EPP), worsened both glycolysis and OXPHOS, along with a decrease in mitochondrial fusion [56], which indicates that common mechanisms may arise in the presence of heme deficiency. To conclude, emerging studies highlight that mitochondrial aberrations may precipitate AIP acute attacks, suggesting that they could be considered a novel therapeutic target.

## 4. AIP Management, Diagnosis, and Standard Treatments

AIP often remains misdiagnosed for more than a decade after the first onset of symptoms and this is possibly explained by the fact that it is a rare disease with low clinical penetrance and manifests with non-specific, episodic symptoms. In certain cases, porphyria is incorrectly diagnosed in patients who do not have porphyria at all. Abdominal pain, usually localized at the epigastric level, alongside autonomic dysfunction, hyponatremia, muscle weakness, or psychiatric symptoms may be key clinical signs of acute porphyria [4]. Patient histories may also aid in diagnosis, especially if the patient experienced appendectomy or cholecystectomy, the incidence of which is common in patients with AIP before their diagnoses [4].

Dark or reddish urine may arouse suspicion of porphyria to the extent that biochemical evaluation of urinary ALA and PBG levels, which are defined as pathologic when they exceed at least 10-fold the normal values, is one of the first approaches to identifying the presence of acute porphyrias. The increase of urinary PBG may direct the diagnosis towards AIP. Therefore, it would be necessary to rule out other types of porphyria, such as VP or HCP, by analyzing porphyrin levels in fecal and blood samples. Nonetheless, the misunderstanding or improper execution of laboratory tests may contribute to the delay in AIP diagnosis. Therefore, even if it is not considered the primary approach, the identification of disease-causing mutations through DNA testing non-invasively collected from the spit or cheek swab is recommended for diagnostic confirmation [62,63]. Genetic screening may be performed in both symptomatic and asymptomatic patients with suspected AIP, and it ranges 90–100% in sensitivity [64,65,66]. Moreover, it can be extended to relatives in order to identify the ancestry of at-risk patients and guide them in the management and care guidelines.

Preventive strategies strongly claim to avoid exposure to precipitating factors and to educate patients with AIP and family members in order to identify them The use of alcohol, smoking and unsafe substances are restricted as well as it is counseled to interrupt porphyrinogen drugs (mostly those inducing CYP-450) and to carry out a healthy diet with an adequate carbohydrate’s intake. For women of child-bearing age (who have a higher incidence of attacks than men), oral contraceptives, progestogen-only pills, implants, and intrauterine devices should be avoided [62].

The current standard of care to manage acute crises in patients with AIP aims to reduce hepatic ALAS1 activity and porphyrin excretion. The gold standard for the treatment of AIP is hemin administration, even if carbohydrate loading is considered in mild cases. LT may represent an alternative option for those patients with high disease activity [7,8]. In 2019, the first liver-targeted strategy based on targeting hepatic *ALAS1* mRNA (givorisan) through a small interfering RNA (siRNA) molecule was approved for the treatment of adult patients with severe AIP by the Food and Drug Administration (FDA) due to the positive results obtained in the phase III ENVISION trial (NCT03338816) [11]. Nonetheless, recurrent attacks may affect patient quality of life and increase the risks of co-morbidities and mortality as well as several shortcomings that emerged with the use of the abovementioned therapies, thereby urging the development of novel, safer therapeutics. Up to now, gene therapy through viral vectors, the newly mRNA-based therapies by which mRNA is incorporated in lipid nanoparticles (LNPs), and the liver-directed recombinant PBGD protein are slowly receiving approval for the treatment of monogenic disorders because they aim to replace the defective proteins and enzymes [15,67]. In addition, the new awareness of AIP pathogenesis is shifting the attention towards the introduction of insulin or insulin-mimetics to ameliorate metabolic alterations [16,17].

### 4.1. Hemin

Intravenous infusion of hemin (Normosang^®^, heme-arginate in Europe; Panhematin^®^, lyophilized hemin in the USA) is a first-line therapy for patients with AIP, especially for those with relapses severe enough to be hospitalized and/or to be admitted to emergency departments [4,68]. Heme directly replenishes the heme pool within the hepatocytes and restores negative feedback on ALAS1 in two ways: (1) it induces transcriptional *ALAS1* downregulation and (2) it hampers ALAS1 enzyme translocation into the mitochondria. Bonkovsky et al. carried out a detailed evaluation of the clinical, biochemical, and genetic features and natural histories of 108 patients with acute porphyrias, including 90 individuals affected by AIP. They revealed that hemin administration was the most effective approach to reduce abdominal pain and other manifestations compared with opiates, non-steroidal anti-inflammatory drugs (NSAIDs), sedatives, and β-blockers. Moreover, repeated Panhematin^®^ infusions resulted in the best option for the prevention of acute episodes to the extent that the authors propose the use of Panhematin^®^ for the prophylaxis of AIP in patients susceptible to recurrent attacks [4].

Nonetheless, the use of prophylactic heme therapy is off-label, and it has been restricted to patients with proven AIP diagnoses and even shows several limitations. Firstly, prolonged heme treatment has not shown relevant benefits to reduce plasma and urinary excretion of porphyrins in patients with AIP, and it may induce chronic liver disease [68,69]. To examine the impact of repeated hemin administration in a condition mimicking severe human AIP, Schmitt et al. treated *Hmbs*^−/−^ mice for eight weeks with PB for three consecutive days and then with hemin for two days. Long-term treatment with hemin promoted chronic hepatic inflammation and oxidative stress in *Hmbs*^−/−^ mice, which further showed paradoxical activation of *AlAS1* and *Heme oxygenase 1* (*HO-1*), the latter of which is involved in heme catabolism [69]. Therefore, if heme infusion effectively reduces *ALAS1* hyperactivity on the one hand, on the other hand its frequent use was associated with the recurrence of attacks as it may decrease the regulatory heme pool, thereby triggering *ALAS1* re-activation and porphyrin accumulation. Accordingly, in a follow-up study that collected the data of 35 patients with AIP, hemin treatment improved acute neurovisceral crisis and the accumulation of heme precursors in all AIP cases, except for eight patients with severe AIP receiving chronic hemin administration who did not show long-lasting clinical and biochemical remission [68]. 

Secondly, several studies reported that patients with AIP with more than four attacks per year receiving prolonged exposure to hemin required port-a-catheter placement. Although the substitution of venous access devices did not represent a limitation for the management of individuals with AIP, it was correlated with thromboembolic events and infections [70].

Finally, multiple heme-arginate injections increased serum ferritin levels and hepatic iron overload as 250 mg of heme-arginate contains ~22.7 mg of iron, thus raising the risk of liver fibrosis [71]. Willandt and collaborators revealed for the first time that monitoring circulating ferritin could be useful for the risk assessment of secondary hemochromatosis and its complications in patients with refractory AIP. Notably, the authors described the case of a female with AIP with recurrent attacks whose ferritin levels rose up to 1904 μg/L after regular heme administration (3 mg/kg/day). At 5-year follow-up, she developed HCC and underwent partial liver resection, in which excessive iron deposition and advanced fibrosis were observed [71]. Afterwards, another follow-up study conducted between 1974 and 2015 in a French cohort that included 602 symptomatic patients with AIP, of whom 46 had regular relapses, showed that the proportion of patients with AIP with recurrent crisis increased from 1995, which coincided with the marketing authorization of hemin [69].

### 4.2. Glucose

The assessment of alimentary regimens in *HMBS*-mutation carriers highlighted that low carbohydrate intake (less than 45–60%) was associated with the risk of malnutrition and disease severity [40,72]. Dietary interventions through carbohydrate loading or intravenous glucose infusions have been proposed for either the treatment or prevention of mild attacks, even if they have not been proven as therapies in controlled trials. Current guidelines for the management of patients with AIP suggest administering 10% dextrose as a temporizing measure if hemin is not available, although monitoring for glycemia is required to avoid additional neurological complications. In the absence of weakness, vomiting, or hyponatremia, high carbohydrate intake 48 h before a specific treatment is recommended [9]. Nonetheless, most patients with AIP manifest these symptoms during the attack, thus failing to tolerate oral polymer solutions of glucose. Therefore, glucose cannot be substitute for hemin to the extent that the latter needs to be introduced if patients with AIP do not show clinical remission within 1–2 days after glucose therapy, or if they experience severe forms of symptomatology [66,73,74].

It was demonstrated that glucose can inhibit hepatic *ALAS1* transcription and porphyrin accumulation by stimulating insulin release. Insulin activates protein kinase B (Akt), which disrupts the interaction between Forkhead Box O1 (FOXO1) and PGC1-α, thereby counteracting the effect of fasting and glucagon [75]. In parallel, glucose may provide substrates for heme biosynthesis. Two molecules of pyruvate derived from glycolysis may be converted in acetyl-CoA by pyruvate dehydrogenase (PDH) and used in the TCA, which produces both reducing equivalents (NADH, FADH2) for OXPHOS and succinyl-CoA [52].

However, the efficacy of glucose therapy has shown unclear experimental and clinical findings for AIP prophylaxis, possibly due to alterations in glucose metabolism observed in both in vitro and in vivo models and in humans [16,17,41,52]. We have recently reported that glucose administration in hepatocytes silenced for *PBGD*-mRNA improved glycolysis but was unable to enhance ATP production [17]. The partial effect of glucose treatment was also examined in AIP mice supplemented with glucose for 10 days before PB-induced attack. AIP mice developing prodromal symptoms were protected against urinary ALA but not PBG accumulation, and were without improved pain and motor disability [16]. Finally, the “glucose effect” may complicate the conditions of patients with AIP. It was observed that the syndrome of inappropriate antidiuretic hormone secretion (SIADH) is a frequent cause of hyponatremia in patients with AIP. Since sodium levels closely depend on blood glucose concentrations, treatment with glucose may produce a pseudo-hyponatremia that makes it difficult to know if it is due to SIADH or if it is secondary to hyperglycemia [76].

### 4.3. Liver Transplantation

Orthotopic LT represents a valid indication for patients with AIP with disabling, unmanageable attacks and refractoriness to hemin treatment [1,10,77]. After PBGD replenishment, urinary and plasma porphyrins fall within the normal range with a significant improvement in neurological and neuropsychiatric manifestations. Conversely, individuals who received LT from symptomatic donors with AIP during a domino LT developed porphyria attacks with elevated porphyrin excretion, abdominal pain, and sensorimotor neuropathy, thus strengthening the notion that the pathophysiology of AIP is strictly dependent on PBGD deficiency in the liver [37].

To date, LT is the only established curative treatment for patients with AIP, which improves their quality of life and prevents the recurrence of the attacks. In a recent case series including 38 patients with AIP, Lissing et al. demonstrated that the survival rate after LT was similar to patients who received LT for other etiologies at 5-year follow-up. The severity of neurological symptoms may influence the survival rate, which was around 92% in patients with AIP with no or moderate neuropathy, and it declined to 83% for those suffering progressive weakness up to wheelchair-bound or bedridden status, and with severe neuropathic pain [78]. Pretransplant renal failure is a common AIP complication and it may hasten the decision to proceed with LT. In some cases, a combined liver-kidney transplant is recommended to preserve renal function [78].

However, long waiting lists due to lack of donors should be taken into consideration as LT is also a highly invasive procedure which may be complicated by high rates of hepatic artery thrombosis and extended periods of immunosuppression [36,79]. In addition, the indication for LT is usually considered late in patients with AIP, many of whom may have developed AIP-related comorbidities over time, indicating that optimal timing of LT may be crucial to reducing the risk of mortality in these subjects [78].

### 4.4. Givosiran: From Preclinical Findings until Now

The use of RNAi, which exploits a biological process by which siRNA selectively targets mRNA expression of a specific gene, has become a promising and clinically validated approach for the treatment of genetically-based disorders. Yasuda and colleagues described the efficacy of a liver-targeted siRNA which inhibits *Alas1* expression (Alas1-siRNA) in AIP mice, thus paving the way for the clinical development of RNAi-based therapies [80]. A panel of 45 siRNAs directed against *Alas1* mRNA were screened to assess their ability to reduce *Alas1* levels and to be delivered into the liver. Among them, the Alas1-siRNA1, which was encapsulated in LNPs, resulted in the most effective induction of robust hepatic *Alas1* inhibition. The injection of Alas1-siRNA1 rapidly lowered ALA and PBG plasma accumulation more than hemin administration after the induction of the acute attack. Alas1-siRNA1 also protected against the development of symptomatology with a duration of ~2 weeks [80].

Afterwards, the same authors developed a novel ALAS1-siRNA conjugated to N-acetyl galactosamine (GalNAc), which binds the liver-specific asialoglycoprotein receptor and contains a target sequence well-conserved across the species (ALN-AS1), with the goal of improving hepatic delivery and finding a method to monitor the duration and kinetics of *ALAS1* inhibition without performing liver biopsy [13,81]. Subcutaneous ALN-AS1 administration in AIP rodents and nonhuman primates strongly attenuated ALAS1 hyperactivity and reduced porphyrin accumulation upon PB-induced attacks. Moreover, hepatic ALAS1 levels matched with those in plasma and urine measured with the circulating extracellular RNA detection (cERD) method, allowing the evaluation of ALAS1 pharmacokinetics through a non-invasive approach which could be translated to patients [79]. Furthermore, the ALN-AS1 escaped the activation of the immune response, with the advantage of being repeatedly administered without inducing the formation of neutralizing antibodies [14].

In 2019, Sardh and collaborators published the results of a phase I randomized trial (NCT02452372) assessing the safety, pharmacokinetics, pharmacodynamics, and outcomes of ALAS1-siRNA-GalNAc subcutaneous administration (givosiran, Alnylam Pharmaceuticals, Inc.) in 40 patients with AIP. The trial consisted of three sections: in part A, patients with ASHE-AIP were randomized to receive one of the five single-ascending doses of givosiran (0.035, 0.10, 0.35, 1.0, or 2.5 mg per kilogram of body weight); in part B, ASHE-AIP patients were assigned to multiple doses of one of two doses of givosiran (0.35 or 1.0 mg per kilogram of body weight); and finally, in part C, givosiran was administered to a subset of patients with AIP with recurrent attacks [82]. Givosiran injections reduced ALAS1 hyperactivity and normalized porphyrin levels in both patients with ASHE and patients with severe AIP, resulting in an acceptable safety profile and lowering the annual attack rate by 79% [82,83]. All eligible patients with AIP in part C were then enrolled in the phase I-II open-label extension (OLE) study, which demonstrated that givosiran maintained a consistent and durable ALA and PBG reduction at both 12 and 18 months, paralleled by a significant decline in annual hemin use [83].

Finally, in the double-blind placebo-controlled trial (ENVISION, NCT03338816), patients with AHP (n = 94, of whom 89 were affected by AIP) with at least two acute episodes were randomized to receive a monthly dose of givosiran or placebo for six months. The annual attack rates were reduced by 74% in patients with AIP compared with the placebo group and were accompanied by sustained low ALA/PBG urinary levels, low daily administration of hemin, and improvement in pain scores, thereby receiving approval by the FDA and the European Medical Agency for the treatment of adult patients with AIP with active disease and adolescent patients aged ≥12 years with AHP [11,12].

Several queries about the use of givosiran remain open. Although it was generally well tolerated, multiple givosiran infusions may cause injection-site reactions, nausea, and fatigue. Hepatic and renal adverse events were reported in 15% of the patients in the givosiran group compared with the placebo group and most of these patients exhibited increased serum creatinine and transaminase levels and reductions in glomerular filtration rates (eGFR). Hyperhomocystinemia (HHcy) is a frequent adverse event which could enhance cardiovascular and neurological risk in symptomatic patients with AIP, especially those receiving heme therapy. Two studies revealed that HHcy and hypermethioninemia accompanied by low blood concentration of pyridoxal-5′-phosphate (PLP) and vitamin B6 occur in patients with AIP even on givosiran regimens [75,84,85]. It has been suggested that heme depletion induced by givosiran may induce the hepatic inactivation of methionine-adenosyl transferase I/III (MATI/III), which is involved in the conversion of methionine into S-adenosylmethionine (SAM), thereby explaining the increased methionine levels. Moreover, givosiran administration may impair heme-dependent proteins involved in Hcy metabolism and clearance, such as cystathionine-β-synthase (CBS) and cystathioninegamma-lyase (CGL). These two enzymes also use PLP and vitamin B6 as cofactors in their physiologic activities, thus offering the opportunities to increase givosiran efficacy by supplementing vitamins. Finally, the main issue concerns both the high cost of the ALAS1-siRNA-GalNAc drug and its long half-life, which could lead to a prolonged inhibition of hepatic heme synthesis, thus influencing heme-dependent liver functions [86].

## 5. Gene Therapy, mRNA-Based Therapies, and Apolipoprotein for Protein/Enzyme deLIVERing: On the Verge of a Scientific Revolution

In 2010, Unzu et al. demonstrated that *PBGD* deficiency in erythroid tissue did not contribute to the acute attacks in AIP mice, whereas their phenotypic manifestations were amenable only to hepatic *PBGD* haploinsufficiency [87]. By overexpressing *PBGD* in the livers of AIP mice through the hydrodynamic injection of a plasmid vector driven by a liver-specific promoter, the authors re-established hepatic PBGD activity paralleled by a great improvement in motor coordination and in biochemical abnormalities after PB-induced attacks [87]. Conversely, AIP rodents receiving bone marrow transplantation from wild-type donors failed to reduce the acute manifestations upon PB administration, indicating that the genetic correction of hepatic PBGD may be promising in the care of patients with AIP. Currently, the best proposals for restoring normal PBGD expression and activity in the liver include (1) gene therapy, (2) mRNA-based strategies, and (3) the new application of apolipoprotein for protein/enzyme deLIVERing.

The first procedure consists of editing the genetic defect by exploiting different technologies such as zinc finger nucleases, CRISPR–Cas9, and engineered, replication-incompetent viral vectors [88,89,90]. Among them, DNA transfer vectors derived from recombinant adeno-associated viruses (rAAV) have been successfully applied in the treatment of inherited disorders and, in recent years, they have been optimized to ensure the long-term expression of a specific protein, to ensure safety and tissue tropism, and to avoid immunogenicity [90,91,92]. The rAAV2 serotype 5 (rAAV2/5) vector, containing human cDNA of the housekeeping *PBGD* isoform, stably sustained hepatic *PBGD* transgene expression and prevented the accumulation of porphyrins during the acute attacks in not only male but also female AIP mice, which are usually less susceptible to rAAV-mediated gene transfer [93]. To maximize the performance of rAAV serotype 5, Pañeda et al. introduced a new cDNA sequence of human PBGD, referred to as human PBGD codon-optimized cDNA (cohPBGD), composed of: (1) a Kozak sequence which enhances translational machinery; (2) enrichment of GC content to improve RNA half-life; and (3), the exchange of rarely used codons with those most frequent in humans. The newly engineered rAAV5-cohPBGD was well tolerated and without side effects in non-human primates (NHPs), showing the highest distribution in the liver compared with the other organs and with a similar efficacy of vector transduction between males and females. In addition, no T-cell responses against AAV capsid proteins or hPBGD were detected in these models after rAAV2/5-cohPBGD injections, thus offering the possibility to be tested in clinical trials [94]. The safety and tolerability profile of rAAV2/5-PBGD administration in eight patients with severe AIP was confirmed in a phase I, open-label, dose-escalation, multicenter clinical trial (NCT02082860). Although patients with AIP improved their quality of life and did not develop cellular responses against PBGD after gene therapy, all of them showed anti-AAV5 neutralizing antibodies against the vector capsid, which blocked adenoviral transduction and had no biochemical remission, probably due to low hepatic PBGD expression.

Therefore, new strategies aiming to improve the efficacy of rAAV-mediated gene therapy are under development [95,96,97,98]. One approach is the substitution of the constitutive promoter of the AAV vector with an inducible one and the modification of the catalytic site of the PBGD. Serrano-Mendioroz et al. introduced two human ALAS Drug-Responsive Enhancing Sequence (ADRES) motifs upstream from the liver-specific promoter of the AAV vector carrying the therapeutic *PBGD* gene. The ADRES enhancers were highly responsive to precipitating factors, thereby providing a high expression of PBGD during an induced attack and taking the advantage to reduce the effective dose by 10-fold [97]. Moreover, by comparing the consensus protein sequence of 12 mammalian species, they identified 2 non-human amino-acid substitutions, isoleucine 291 to methionine (I291M) and asparagine 340 to serine (N340S), which improve PBGD stability and generate a hyper-functional isoform conferring full protection on AIP mice against PB-induced attacks [98].

The mRNA-based therapies are a new class of drugs which represent a promising treatment for genetic diseases, with possible applications even in other fields, such as cardiology, infectious disease, and oncology. The mRNA is packaged in LNPs and delivered into specific tissues through endocytosis. LNPs have been designed to interact with the low-density lipoprotein receptor (LDLR) expressed on the hepatocyte surface and to escape from phagocytosis through a polyethylene-glycol (PEG) coating, thereby protecting the mRNA content from nuclease degradation and immunity [13,14,67]. Then, LNPs are degraded by lysosomes, whereas the mRNA is released into the cytoplasm and translated in the ribosomal machinery into a specific protein. Differently from protein/enzyme replacement therapy, which limits its use to the replacement of secreted proteins or peptides and which needs a protein-protein or protein-receptor interaction to produce a response, the mRNA-based delivery system offers the possibility to synthetize a protein or enzyme which could localize in any type of subcellular compartment, allowing its natural interactions and post-translational modifications to occur. Moreover, the injections of LNP-encapsuled mRNA avoid genotoxicity as they induce a transient protein replacement, but they may need to be repeatedly administered. Human PBGD (hPBGD) mRNA administration in AIP mice rescued the hepatic PBGD activity over the course of the acute attack (~7–10 days) and its stability in hepatic parenchyma was enhanced when PBGD mRNA was conjugated with liver-targeted apolipoprotein A1 (ApoA1), a structural part of high-density lipoprotein (HDL) [15,99,100]. A sustained PBGD genetic correction reduces ALA/PBG accumulation along with alleviation of pain, motor discoordination, and disturbances in nerve conduction velocity. Moreover, multiple cycles of hPBGD mRNA injections normalized urinary porphyrin levels in mice, rabbit, and NHPs, which did not show activation of the immune response or aberrations in liver function tests, indicating they are a promising candidate for translatability to humans [67,99,100].

Finally, an innovative strategy for the treatment of AIP was recently proposed by Còrdoba et al., who developed a recombinant PBGD protein by binding the ApoA1 to the N-terminal of the human PBGD (rhApoA1-PBGD) or to the hyper-functional PBGD carrying the I291M and N340S amino-acid substitutions (rhApoA1-PBGDms). The ApoA1 exploits the centripetal transport of HDL from periphery to the liver and it is internalized into hepatocytes by the ApoA1 receptors. Both rhApoA1-PBGD and rhApoA1-PBGDms restored PBGD activity and heme biosynthesis not only in the liver but also in brain tissue, showing the ability to cross the blood–brain barrier (BBB). Notably, while the rhApoA1-PBGD prevented the accumulation of ALA and PBG after PB-induced attack, the rhApoA1-PBGDms provided long-lasting protection against multiple attacks, thereby resulting appealing for the management of patients with recurrent episodes [15].

## 6. Insulin and Insulin-Mimetics: Possible Alternative Strategies for Hereditary Metabolic Disorders?

In the previous sections, we reported the most recent findings highlighting alterations in glucose metabolism and hormonal regulation linked to defects in heme production. The current knowledge of AIP pathogenesis and its metabolic implications have supported the hypothesis that the efficacy of carbohydrate loading for the treatment of mild AIP cases could be limited by the glucose intolerance or hyperinsulinemia observed in patients with AIP [16,38,72]. Oral or intravenous carbohydrate surplus, leading to an increment in insulin levels, have been associated with low biochemical disease activity in a case-control study including 50 patients with AIP, probably due to phosphatidylinositol 3-kinase (PI3K)/Akt-mediated ALAS1 inhibition [72,75]. Accordingly, sustained hyperinsulinemia, excess body weight, or the presence of type 2 diabetes mellitus (T2DM) in individuals with AIP featured in those patients with stable disease [16,38].

Recently, it was observed that therapeutic interventions aimed at increasing circulating HDL and ApoA1 levels improved hyperglycemia and attenuated IR in db/db mice, a genetic model of T2DM [101], thereby opening the possibility to consider insulin or insulin-sensitizers to correct metabolic dysfunctions by improving insulin sensitivity and enhancing glucose therapy in the AIP context. By comparing the efficacy between a fast-acting insulin (Actrapid^®^) and a liver-targeted insulin fused to ApoA1 (Ins–ApoA1), both co-administered with glucose, Solares et al. found that both exogenous insulins injected into AIP mice normalized heme biosynthesis, gluconeogenesis, and increased the number of mitochondria and the levels of circulating triglycerides, probably to promote the synthesis of high-energy molecules. Nonetheless, Actrapid^®^ did not hamper urinary ALA or PBG excretion after PB challenge, while Ins–ApoA1 sustained hepatic *Alas1* downregulation accompanied by porphyrin reduction. In addition, prophylactic treatment with the Ins–ApoA1 and glucose mixture showed a significant improvement in pain scores and motor coordination in AIP mice, although high ALA/PBG accumulation was maintained [16].

Our group recently carried out a proof-of-concept study by exploiting alpha-lipoic acid (α-LA), an insulin-mimetic already used as a nutraceutical approach to the treatment of T2DM and hepatic steatosis, in an in vitro model of AIP. An endogenous cofactor of α-ketoglutarate, α-LA offers the advantage of stimulating the TCA and producing substrates for heme biosynthesis. HepG2 cells were silenced for the *PBGD* mRNA (siPBGD) and cultured in a medium with low glucose content to mimic fasting. Administration of α-LA, either alone or in combination with glucose (α-LA+Gluc), increased the heme content in hepatocytes, probably by stimulating PBGD expression and activity [17]. Moreover, α-LA strongly enhanced ATP production and triglyceride secretion even more when combined with glucose, resembling the action of the exogenous insulins used by Solares et al. [16,17]. Interestingly, we revealed that the beneficial effects of co-administration of α-LA+Gluc may be due to the re-establishment of the glycolytic-mitochondrial respiratory network. This effect may be ascribed to a series of events that may occur in siPBGD hepatocytes treated with both α-LA and glucose: if glucose activates glycolysis on the one hand, then on the other hand, α-LA rescues heme biosynthesis by boosting the TCA and, consequently, the production of cytochromes, thereby improving OXPHOS activity and supplying cells with energy fuels. The restoration of hepatocellular homeostasis was even coupled to the recovery of mitochondrial dynamics, in terms of fusion, fission, and overall mitochondrial mass [17].

Although the long-term efficacy of α-LA needs to be evaluated in future studies and in humans, promising results regarding its safety and tolerability have already been achieved in patients with metabolic disorders and in those affected by PCT [18,102,103].

## 7. Conclusions

Ongoing efforts for the deeper understating of the pathogenesis, management, and treatment of AHPs are gradually getting closer to resolution. Many studies focused on AIP as it represents the most disabling and frequent type among the AHPs, with a high variability in both causative mutations affecting the *HMBS-*gene and sub-clinical phenotypes. Substantial evidence has established that the prevalence of *HMBS* genetic mutations is relatively high in the general population, but its penetrance may depend on the degree of kinship and genetic and environmental factors, which may even contribute to the delay in AIP diagnosis or misdiagnosis.

Glucose for mild to moderate cases and hemin for recurrent crises, both modulating hepatic *ALAS1* expression, remain the treatments of choice, although they both have shown several drawbacks. Currently, LT represents the only curative option, but it is reserved for patients with AIP with advanced disease and refractoriness to hemin infusions. To date, givosiran, the ALAS1 siRNA inhibitor, has been approved for the management of AIP in adults and adolescents and shows interesting results in reducing annual attack rates and improving pain scores. However, hepatic and renal adverse events as well as possible side effects in the long-term due to the persistent inhibition of heme biosynthesis must be taken into consideration.

Additional strategies are emerging and are currently under development for AIP treatment. Gene therapy through AAV vectors, liver-targeted mRNA-based approaches exploiting LNPs, and recombinant human PBGD combined with ApoA1 displayed well-tolerated and safe profiles in small and large experimental models of AIP, showing high hepatic tropism, little immunogenicity, and significant improvements in biochemical alterations, pain, and motor disability. Optimization of these procedures through the monitoring of pharmacokinetics and pharmacodynamics, modifications to the human PBGD amino-acid sequence, the control of the immune response, and the assessment of the procedures’ effects after multiple administrations are still ongoing, even as the positive results achieved over the past few years are raising hopes for a definite cure.

Finally, new awareness of the metabolic profiles of patients with AIP, mostly characterized by alterations in glucose metabolism, mitochondrial dysfunction, and metabolic reprogramming, is paving the way for the introduction of liver-targeted insulin and/or its mimicking agents in order to improve the efficacy of glucose therapy. Although this field is still at the dawn, clinical evaluations of liver metabolism and metabolic signatures combined with insulin sensitizers for the treatment of patients with AIP presenting with metabolic disturbances such as glucose intolerance, IR, obesity, and T2DM may potentially sustain the hepatic absorption and maintenance of glucose homeostasis from dietary carbohydrates, thereby improving liver function. A representative scheme of the current and ongoing therapies is illustrated in Figure 2.

## Figures and Tables

**Figure 1 biomedicines-10-00648-f001:**
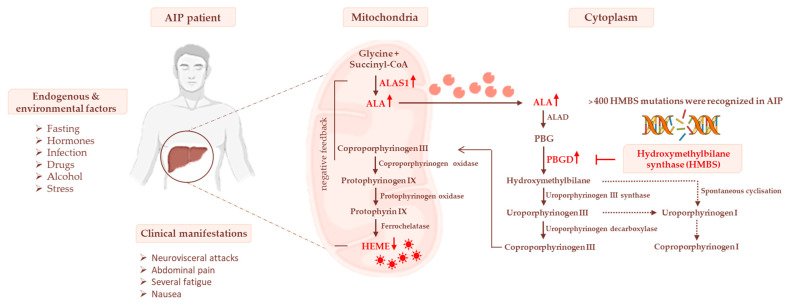
The heme biosynthetic pathway and AIP pathophysiology. Heme biosynthesis begins in mitochondria with the conversion of glycine and succinyl-CoA to δ-aminolevulinic acid (ALA) through the δ-aminolevulinic acid synthase-1 (ALAS1) enzyme. In the cytoplasm, ALA is metabolized to porphobilinogen (PBG) and then to COPROgen III. The latter is then transported into mitochondria for the synthesis of heme, which in turn downregulates ALAS1. The deficiency of HMBS genes causes a reduction in hepatic heme synthesis, leading to the stopping of its negative feedback on ALAS1. Thereby, the levels of porphyrin precursors (ALA and PBG), mainly in response to precipitating factors, may accumulate in the liver, systemic circulation, and urine, triggering neurological damage.

**Figure 2 biomedicines-10-00648-f002:**
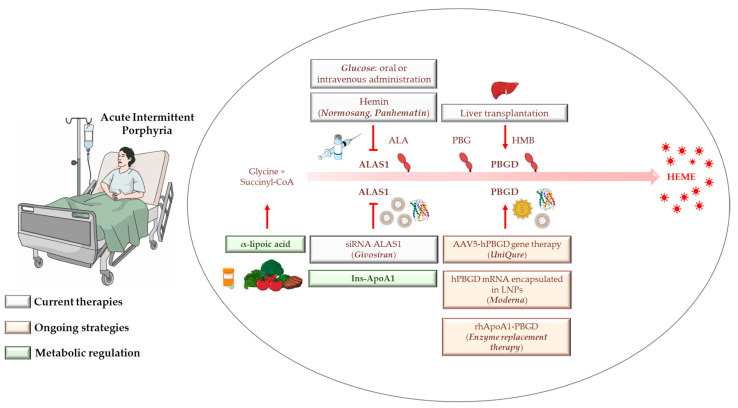
A schematic overview of the current and ongoing efforts in the treatment of AIP and their therapeutical targets. In the white boxes, hemin, glucose, liver transplantation, and the recently approved givosiran represent the standard treatments for AIP management. Therapies under development are in peach-pink and include AAV5-hPBGD gene therapy, hPBGD-mRNA delivered by LNPs, and liver-targeted rhApoA1-PBGD. Finally, special attention should be given to insulin and insulin-sensitizers (light green) as potential alternative strategies for the correction of metabolic dysfunction and the enhancement of glucose therapy.

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
