# Peer review of "Cutting-Edge Therapies and Novel Strategies for Acute Intermittent Porphyria: Step-by-Step towards the Solution"

_biomedicines, 2022, doi:10.3390/biomedicines10030648_

Round 1

Reviewer 1 Report

The authors make a very good compendium of data regarding the novel strategies for AIP, a topic of great interest in the last years in the field of rare diseases; it is a well structured and complete review.

I have only two specific comments:

  1. in the section 4.1 (Hemin) I think it is important to better specify that prophylactic administration of hemin is off-label, although used in clinical practice
  2. in the section 4.4 (Givosiran: from preclinical findings up to now), at the end where the authors report adverse events (ISR, hepatic and renal), I think that also hyperhomocysteinemia (and management with vitamin supplementation) should be reported 

Author Response

We sincerely thank the Reviewer for the positive feedback about the manuscript. According to her/him, we improved the review with information about the prophylactic hemin use (Yellow Tracks in the section 4.1) and we expanded the chapter “Givosiran: from preclinical findings up to now” by discussing about hyperhomocysteinemia and vitamin supplementation (Yellow Tracks in the section 4.4).

Reviewer 2 Report

Well written manuscript from the known experts in the topic. I have few minor comments that the authors may want to consider. Please see the attached file.

I congratulate the authors for such a wonderful work!

Author Response

We thank the Reviewer for the positive comments. As suggested by her/him in Point 1, we added the population-based study carried out by Lithner et al. about the beneficial effects of diabetes in AIP subjects. It has been demonstrated that at the onset of diabetes mellitus, AIP patients ceased suffering acute attacks and decreased ALA/PBG urinary excretion. Surprisingly, the presence of diabetes in AIP individuals was not associated with the development of HCC. On the other hand, it has been described that diabetes may activate hepatic ALAS1 in mice and a case report revealed that the concomitant overt of diabetes mellitus in one AIP subject was paralleled by the late manifestation of acute episodes, supporting that the impact of IR in AIP pathology needs to be further elucidated (Yellow Tracks in the section 3.1. “The impaired glucose homeostasis and IR contribute to AIP acute attacks”).

As concerns Point 2, few studies have reported a correlation between microbiome and porphyrin metabolism. For instance, intestinal Bacteroides depends on porphyrins as protoporphyrin IX or iron-charged heme for their normal growth, while Escherichia coli, commonly presents in the intestinal flora, is a porphyrin donor supporting a possible cross-interaction between the two species. Porphyrins, especially photosensitized ones, have a potent antiviral and antimicrobial activity to the extent that they may induce DNA damage in several pathogenic strains as Staphylococcus aureus and Escherichia coli. Conversely, it has been reported that dietary heme may be metabolized by sulfide-producing bacteria and mucin-degrading bacteria, which may promote gut permeability and epithelial hyperproliferation thus increasing the risk of colorectal cancer.

Although the issue arisen by the Reviewer about the microbiome and the porphyria is intriguing, articles describing bacterial signature in AIP subjects and, if it could be modulated among patients with stable disease, highly excreters and symptomatic ones, are still missing.

Reviewer 3 Report

The review is very interesting, usefull and well organized.  I appreciated also the figures. Congratulations!

Author Response

We thank the Reviewer for the positive comments.